# The Relationship between Vitamin D Status and the Menstrual Cycle in Young Women: A Preliminary Study

**DOI:** 10.3390/nu10111729

**Published:** 2018-11-11

**Authors:** Karolina Łagowska

**Affiliations:** Poznan University of Life Sciences, Institute of Human Nutrition and Dietetics, Dietetic Division, ul. Wojska Polskiego 28, 60-637 Poznań, Poland; karolina.lagowska@up.poznan.pl; Tel.: +48-61-8487335

**Keywords:** menstrual cycle, vitamin D, 25(OH)D, nutritional status

## Abstract

Background: The aim of this study was to evaluate serum vitamin D levels and to compare these with the menstrual cycle in young women with different body weights. Methods: Eighty-four students were recruited into the study of which 77 remained at the study’s completion. Women were assigned to one of two subgroups, according to their 25-hydroxy vitamin D test level [25(OH)D] in which 60 women had low 25(OH)D levels (LD < 30 ng/mL) and 17 had normal levels (ND > 30 ng/mL ≤ 80 ng/mL). Results: In the LD group, 40% of participants reported having long cycles, 27% were classified as having oligomenorrhoea, and 13% as having amenorrhoea. In the ND group, only 12% reported menstrual cycle disorders, 6% had oligomenorrhoea, and 6% had amenorrhoea. Women who did not meet the recommended level of 30 ng/mL of 25(OH)D had almost five times the odds of having menstrual cycle disorders as women who were above the recommended vitamin D level. Conclusion: A relationship was demonstrated between the frequency of menstrual disorders and low levels of vitamin D. Supplementation is necessary in women with low levels of vitamin D in order to compensate for this deficiency and to assess its effect in regulating menstrual disorders.

## 1. Introduction

Vitamin D deficiency, which arises due to insufficient exposure to sunlight and dermal synthesis from 7-dehydrocholesterol and limited intake from food and supplements, disrupts the function of all systems of the body and increases the risk of osteoporosis, cancer, cardiovascular disease, autoimmune disease, and mental disorders such as depression and chronic pain syndrome [1,2,3].

In humans, vitamin D_3_ acts via the vitamin D receptor (VDR) to modulate the expression of approximately 3000 genes in various tissues including reproductive tissues such as the ovaries, uterus, and vagina [4]. Furthermore, genetic polymorphisms related to vitamin D receptors have been linked to serum levels of luteinizing hormone, sex hormone-binding globulin, testosterone, and insulin [5,6]. This active, new area of investigation has attracted more interest recently with many studies reporting a relationship between low serum 25-hydroxyvitamin D (25(OH)D) levels and the symptoms of insulin resistance, hirsutism, and infertility are associated with ovulatory disorders, which can lead to sterility and polycystic ovary syndrome (PCOS). In addition, it has been postulated that increased serum vitamin D and calcium levels may improve reproductive function in women with PCOS. However, population-based human data on vitamin D and menstrual cycles are still lacking.

Our study aimed to examine the association of vitamin D with menstrual cycle characteristics including amenorrhoea, oligomenorrhoea, and eumenorrhoea in young women with different body weight but without PCOS.

## 2. Materials and Methods

In April, 84women were recruited from three Poznań universities to the study of whom 77 remained at the study’s completion. The exclusion criteria were serious medical conditions, use of hormonal contraception or other medications that might interfere with the hypothalamic–pituitary gonadal axis, use of drugs such as psychotropics that could increase weight and interruptthe menstrual cycle, use of vitamin D supplements in the last six months, clinical diagnosis of eating disorders, and history of clinical diagnosis of primary ovarian failure, hyperprolactinemia, thyroid dysfunction or polycystic ovary syndrome hirsutism or acne suggestive of hyperandrogenism or PCOS. 

Written informed consent was obtained from all participants. The study was approved by the Poznań Medical Ethics Committee (No. 868/15). 

Each subject completed a medical questionnaire. The questions concerned menstruation. The questions focused on age at menarche, length of the menstrual cycles, and history of menstrual disorders.

Primary amenorrhoea was diagnosed where there was no onset of menses by the age of 15 while secondary amenorrhoea was diagnosed when there was no menstruation for six months or for longer than three times the previous cycle length. Menstrual periods that occurred more than 35 days apart were described as oligomenorrhoea [7]. 

Blood samples were obtained from menstruating subjects between days 2 and 5 of the menstrual cycle (in the early follicular phase) and at random in amenorrheic subjects. Blood samples were taken between 6:00 a.m. and 9:00 a.m. following overnight fasting and rest. The women were instructed to abstain from caffeine and alcohol for 24 h prior to the blood sampling and to refrain from strenuous exercise on the day of sampling. 

Concentrations of vitamin D as 25(OH)D were obtained by using liquid chromatography tandem mass spectrometry. Samples were determined in duplicate.

Hirsutism was graded according to the standard Ferriman–Gallwey score by which the density of terminal hairs is scored at nine different body sites [8,9]. In each of these areas, a score of 0–4 was assigned. A total score of over 8 was interpreted to mean that hirsutism was present. Acne was graded according to the Pillsbury method, which counts acne lesions and ranks their severity into grade I (absent or minor, 1–9 comedones), grade II (mild, 10–19 comedones), grade III (moderate, R20 comedones, inflammation), and grade IV (severe) [8,10]. Seven of the subjects presented hirsutism, acne, alopecia, or voice deepening and were, thus, excluded from the study. 

Seventy-seven women were assigned to one of the two subgroups according to their 25(OH)D concentration: 60 women had low 25(OH)D concentration (LD < 30 ng/mL, LD group) and 17 women had normal 25(OH)D concentration (ND > 30ng/mL, ≤80 ng/mL, ND group).

In addition, women were also grouped according to the length of their menstrual cycle. One group consisted of those with cycle disorders (oligomenorrhoea or amenorrhoea, MD, *n* = 26) while the second was composed of those who menstruated regularly (RC, *n* = 51). 

In order to evaluate the nutritional status of the subjects, anthropometrical indices, height, and weight were measured by using an anthropometer coupled with a WPT 200 OC verified medical scale (Rad Wag). Body mass index BMI (kg/m^2^) was calculated as body weight divided by body height. 

The participants were dressed in minimal clothing during the measurements, which were rounded to the nearest 0.5 kg and 0.5 cm. Body fat mass (FM) and fat-free mass (FFM) were determined in the morning following an overnight fast using a Bod Pod analyzer with subjects lying in a supine position. 

Means and standard deviations of the quantitative variables were calculated. Normality of the distribution was determined. Means were compared between the subgroups, according to different concentrations of vitamin D (LD and ND) and the menstrual cycle type (MD and RC) using analysis of variance when the distribution was normal and the Kruskal–Wallis test if continuous variables were skewed. The odds ratios for menstrual disorders for women with low vitamin D levels were also calculated.

The statistical analysis was performed by using *Statistica 13.0* software (StatSoft, Palo Alto, CA, USA). *P*-values of less than 0.05 were considered statistically significant. 

## 3. Results

Most women (79%, *n* = 60) had low vitamin D concentrations including (25(OH)D concentrations below 30 ng/mL) and almost 65% of this LD group had insufficient vitamin D concentration (25(OH)D concentration below 20 ng/mL). The mean 25(OH)D concentration was 18.59 ± 5.4 ng/mL in the LD group and 35.74 ± 2.98 ng/mL in the ND group. In the LD group, 40% of participants reported menstrual disorders, 27% were classified as having oligomenorrhoea, and 13% as having amenorrhoea. In the ND group, only two subjects reported menstrual cycle disorders of which one had oligomenorrhoea and one had amenorrhoea. In the LD group, almost 37% of women were overweight or obese while the ND group had only one person (6%) who was obese. Women from the LD group were characterized by significantly higher levels of adipose tissue and larger waist circumferences (Table 1).

Lower concentrations of 25(OH)D were associated with long cycles (oligomenorrhoea or amenorrhoea). Women who were below the recommendation of 30 ng/mL of 25(OH)D had almost five times the odds of having disorders in menstrual cycles than did women who were above 30 ng/mL (OR(CI): 5.0 (1.047 to 23.871), *p* = 0.04) (Table 2).

Women from the MD group were characterized by higher body mass, higher BMI, higher body fat content, and greater waist circumference as well as by lower concentrations of vitamin D. Most of the women (58%) in the MD group had insufficient vitamin D concentrations while, in the RC group, 37% of women had vitamin D concentrations below 20 ng/mL. In the MD group, only two individuals (7%) had normal vitamin D concentration while, in the RC group, 19 women (37%) had sufficiently high levels (Table 3).

## 4. Discussion

The existence of vitamin D deficiencies in the general population has been demonstrated many times in previous studies [11]. In the work of Karczmarewicz et al. [12], up to 80% of the population was found to be deficient. In the work of Jukic et al. [13], as many as 76% of women (484 out of 606) had vitamin D concentrations below 20 ng/ml, which is a very similar result to the one obtained in this study where 25(OH)D deficiency was found in 78% of women. In addition, the anthropometric parameters and body composition measured in this study are similar to the values described by other authors where low concentrations of vitamin D were significantly associated with higher body weight and BMI, which indicates being overweight or obese, greater adipose tissue, and larger waist circumferences [14].

The main aim of this study was to assess the prevalence of menstrual disorders in women with different body weights according to their vitamin D concentration.

It has been reported that low vitamin D concentrations co-occur with disturbed menstrual cycles [13]. In this study, it has been found that women with oligomenorrhoea and amenorrhoea are characterized by significantly lower vitamin D concentration than women with regular cycles. In addition, in our study, lower plasma levels of 25(OH)D were associated with increased odds of having menstrual disorders (oligomenorrhoea or amenorrhoea). Similar results were found in the study of Sadhir et al. [15]. The study of Jukic et al. [13] is one of the few to have examined the 25(OH)D concentration and the length of the menstrual cycle in women of childbearing age. They found that a lower 25(OH)D concentration correlated with an irregular menstrual cycle but not with short (<21 days) or long cycles (>32 days).

Additionally, the work of other authors has shown that a deficiency in vitamin D can lead to an increase in the parathyroid hormone, which is accompanied by PCOS, infertility due to lack of ovulation, and high testosterone levels. Vitamin D controls the biosynthesis of estrogen by directly regulating the aromatase gene and by maintaining extracellular calcium homeostasis. Vitamin D also has a significant effect on the action of insulin, which impacts the presence of the VDR receptors in pancreatic β-cells to which calcitriol binds and stimulates insulin secretion [16]. It also participates in calcium metabolism. Vitamin D deficiency, with the additional dysregulation of calcium metabolism in the body, contributes to the suppression of ovarian follicular maturation in women with PCOS. There are also studies showing that vitamin D supplementation can regulate the course of the menstrual cycle in women with PCOS [11,15]. However, there are too few studies evaluating the effects of supplementation as a means of regulating menstrual periods to allow unambiguous conclusions to be drawn about optimal dosing.

Answers are needed to the question of whether a similar mechanism occurs in women whose menstrual disorders are not related to PCOS. The women recruited to this study had never been diagnosed with PCOS and, apart from their increased body weight, they did not show other physical features typical of this disease. To this end, patients were removed from the study if physical features such as hirsutism, alopecia, or acne were present, which suggests undiagnosed hyperandrogenism or PCOS. To the best of our knowledge, this is the first study to find lower levels of vitamin D in young women who are 18 to 25 years of age with menstrual disorders but without PCOS. Only one study has been published in which menstrual disorders were found in women without PCOS, but this was a female population of late-reproductive age [13].

There are studies suggesting that menstrual disorders in women without PCOS may be due to vitamin D’s effect on the synthesis of anti-Müllerian hormone (AM), which is a marker of ovarian reserve. It has been shown that vitamin D supplementation prevents seasonal drops in AMH levels in women. In human granule cells, vitamin D_3_ lowers the expression of the receptors for AMH (AMHR-II) and FSH (FSHR). It also alters the secretion of AMH in the granulosa cells of the ovarian follicles and decreases their sensitivity to FSH, which may play an important role in the development of the follicles [17]. Research suggests that low levels of vitamin D are correlated with low levels of AMH, which is also associated with a decrease in the ovarian reserve and an increase in FSH at an early stage of the cycle and, thus, is a significant reduction in fertility [15,17,18].

Our study has a number of limitations. Menstrual cycles were evaluated using a questionnaire. For a full diagnosis of menstrual cycle disorders, it would be necessary to determine hormone levels and for a gynecologistto perform an ultrasound examination. Accurate diagnosis of menstrual cycle disorders would make it possible to determine their etiology (including pituitary insufficiency and thyroid disorders). Studies of the relationship between vitamin D status and the menstrual cycle indicate that several confounding factors may affect the results. Although these factors include stress, these studies failed to estimate the levels of mental and physical stress in the study population. Large-scale randomized controlled trials are needed to confirm our results and to evaluate the effects of vitamin D supplementation on treating menstrual disorders. It is important to use the optimal supplement at the dose of vitamin D. Toxic dose of vitamin D has not been established. An Institute of Medicine report concluded that doses below 10,000 IU/day are not usually associated with toxicity while doses equal to or above 50,000 IU/day for several weeks or months are frequently associated with toxic side effects including documented hypercalcemia. Most reports of vitamin D toxicity have involved vitamin D intake of over 40,000 IU/day [19,20,21]. The clinical manifestations of vitamin D toxicity are a consequence of hypercalcemia and include fatigue, generalized weakness, anorexia, polyuria, polydipsia, dehydration, constipation, nausea, vomiting, confusion, difficulty concentrating, irritability, drowsiness, and coma [19,20,21].

## 5. Conclusions

Overall, this study can be considered preliminary in nature. The study demonstrates a relationship between menstrual disorders and low levels of vitamin D, which may be related to AMH, insulin, and androgen or which could perhaps involve a yet-to-be-identified pathway. Further investigation of potential mechanisms is needed. Supplementation is recommended in women with low levels of vitamin D in order to compensate for this deficiency and to assess its effect in regulating menstrual disorders.

## Figures and Tables

**Table 1 nutrients-10-01729-t001:** Characterization of anthropometric parameters, body composition, and menstrual cycle of participants.

	25(OH)D < 30 ng/mL (*n* = 60)	25(OH)D > 30 ng/mL (*n* = 17)
Age (years)	21.6 ± 2.4	22.1 ± 1.9
Height (cm)	166.4 ± 5.4	165.6 ± 3.8
Body mass (kg)	66.7 ± 12.8	57.0 ± 9.5 **
FM (%) <26 >26	29.7 ± 6.9 21 (35) 39 (65)	25.1 ± 5.4 * 10 (63) 6 (37)
Regular cycle RC (*n*, %)	36 (60)	15 (88)
Menstrual disorders MD (*n*, %) Oligomenorrhoea (*n*, %) Amenorrhoea (*n*, %)	24 (40) 16 (27) 8 (13)	2 (12) 1 (6) 1 (6)
BMI (kg/m^2^) <18.9 19–24.9 25–29.9 >30.0	24.1 ± 4.6 4 (7) 34 (57) 15 (25) 7 (11)	20.8 ± 3.0 * 4 (25) 11 (69) - 1 (6)
Waist circumference (cm)	112 ± 14.1	73 ± 11.2 **
Vitamin D (ng/mL) (*n*, %) <20 20–30 30–80 >80	18.6 ± 5.4 39 (65) 21 (35) 00	35. ± 2.3 ** 0 0 17 (100) 0

* *p* < 0.05, ** *p* < 0.001, *** *p* < 0.0001; FM, Body fat mass.

**Table 2 nutrients-10-01729-t002:** Association of 25-hydroxyvitamin D with menstrual cycle characteristics among study subjects.

Odds ratio (LD group/ND group)	5.000
95% CI	1.047 to 23.871
*Z* statistic	2.018
Significance level	*p* = 0.044 *

* *p* < 0.05.

**Table 3 nutrients-10-01729-t003:** Characterization of anthropometric parameters, body composition, and concentration of vitamin D in study participants with and without menstrual disorders.

	Menstrual Disorders (*n* = 26)	Regular Cycles (*n* = 51)
Age (years)	21.3 ± 2.3	22.0 ± 2.7
Height (cm)	167.0 ± 5.1	166.0 ± 4.5
Body mass (kg)	77.6 ± 12.1	57.9 ± 10.5 **
FM (%) <26 >26	34.5 ± 6.1 2 (7) 24 (93)	25.7 ± 5.9 ** 29 (57) 22 (53)
BMI (kg/m^2^) <18.9 19–24.9 25–29.9 >30.0	27.9 ± 4.2 1 (4) 4 (15) 13 (50) 8 (31)	21.1 ± 4.9 ** 8 (16) 41 (80) 2 (4) 0
Waist circumference (cm)	106 ± 13.1	79 ± 12.2 **
Vitamin D (ng/ml) (*n*, %) <20 20–30 30–80 >80	18.5 ± 4.4 15 (58) 9 (35) 2 (7) 0	24.9 ± 3.1 * 19 (37) 13 (26) 19 (37) 0

* *p* < 0.05, ** *p* < 0.001, *** *p* < 0.0001; FM, Body fat mass.

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
