# Peer review of "The Relationship between Vitamin D Status and the Menstrual Cycle in Young Women: A Preliminary Study"

_nutrients, 2018, doi:10.3390/nu10111729_

Round 1

Reviewer 1 Report

There are two big problems with this study. The results suffer from A) confounding by condition   (the recruits may, for instance have been taking drugs such as psychotropics that both increased weight and interrupted menstrual cycle and from B) question about direction of effects as in all correlation studies i.e. what has caused what.

How were the women recruited, from where? What are their ages? How was the sample size calculated? Were the women and the investigators blind to the hypothesis? Was drug taking, diet, sun exposure documented and controlled? What about initial weight? Stress levels?

Where are the references for Vit D deficiency causing cancer, cardiovascular problems?

Limitations need substantial expansion.

Thee should be a word of caution re Vit D overdose.

Author Response

Enclosed, please find the revised version of manuscript entitled "The relationship between vitamin D status and the menstrual cycle in young women: A preliminary study" by Karolina Łagowska, manuscript ID: 381336 . I appreciate your constructive and helpful comments and those of the reviewers. I feel that my manuscript is now greatly improved.

1. There are two big problems with this study. The results suffer from :

A) confounding by condition  (the recruits may, for instance have been taking drugs such as psychotropics that both increased weight and interrupted menstrual cycle

Answer: The subjects recruited to the study didn't taking a drugs such as psychotropics or different which could increased weight and interrupted menstrual cycle (such as hormonal contraception). Authors added those information's in the text to the inclusion criteria's (page 2, line 44-49).

B) and from question about direction of effects as in all correlation studies i.e. what has caused what.

Answer:

In all correlation or observational studies we  have a problem with what has caused what. In this study we formulated close inclusion criteria (line 44-50) to eliminated the mistakes.

2. How were the women recruited, from where? What are their ages? How was the sample size calculated? Were the women and the investigators blind to the hypothesis? Was drug taking, diet, sun exposure documented and controlled? What about initial weight? Stress levels?

Answer:

Women were recruited to the study from Poznan Universites (page 2, line 43). What is more, in the table 1 Author gave the dates about age and body weight of Participants (line 106-108). The inclusion criteria were no serious medical conditions, no use of hormonal contraception or other medications that might interfere with the hypothalamic–pituitary–gonadal axis, drugs such as psychotropics that both increased weight and interrupted menstrual cycle, no use of vitamin D supplements in the last 6 months, no clinical diagnosis of eating disorders, no history of clinical diagnosis of primary ovarian failure, hyperprolactinemia, thyroid dysfunction or polycystic ovary syndrome, and not smoking (page 2, line 44-49). Sample size was estimated using G-power program. Effect size d = 0.8 and primary outcomes like differences in 25(OH)D level, anthropometric parameters or body composition. This study was estimated in April, after the winter when the insolation in Poland is very low. Author added this information's in the text (line 43).

3. Where are the references for Vit D deficiency causing cancer, cardiovascular problems?

Answer:

Author didn't add information's about Vit D deficiency causing cancer, cardiovascular problems because the aim of this study was estiamtion the relationship between vitamin D status and the menstrual cycle in young women. Author of this study wanted focused only on the most important issues connected with menstrual cycle. Relationship between vit D deficiency, cancer and cardiovascular problems are very good known and many authors mentioned about it earlier.

4. Limitations need substantial expansion.

Answer:

Author added information's about limitations in the discussion section (page 5, line 179-184).

5. There should be a word of caution re Vit D overdose.

Answer:

Author added information's about vitamin D overdose in the discussion section (page 5, line 184-192).

Reviewer 2 Report

Please correct the manuscript according to the suggestions mentioned in the pdf.

Author Response

Enclosed, please find the revised version of manuscript entitled "The relationship between vitamin D status and the menstrual cycle in young women: A preliminary study" by Karolina Łagowska, manuscript ID: 381336 . I appreciate your constructive and helpful comments and those of the reviewers. I feel that my manuscript is now greatly improved.

Line 95 and 96

Answer: According to the reviewer suggestion I removed the bracket.

Line 149

Answer: According to the reviewer suggestion I gave the refferences

Round 2

Reviewer 1 Report

see the attached file.

Author Response

Enclosed, please find the revised version of manuscript entitled "The relationship between vitamin D status and the menstrual cycle in young women: A preliminary study" by Karolina Łagowska, manuscript ID: 381336 . I appreciate your all editors comments. I feel that my manuscript is now greatly improved.

Reviewer 2 Report

Both reference 21 and 22 does not conform that the vitamin D can increase the insulin secretion.

remove the sentence if u cant give the reference.

Author Response

Enclosed, please find the revised version of manuscript entitled "The relationship between vitamin D status and the menstrual cycle in young women: A preliminary study" by Karolina Łagowska, manuscript ID: 381336 . I appreciate your constructive and helpful comments and those of the reviewers. I feel that my manuscript is now greatly improved. My manuscripts was also checked by native speaker.

Reviewers:

Both reference 21 and 22 does not conform that the vitamin D can increase the insulin secretion. Remove the sentence if u can't give the references.

Answer:

I changed the references 21 and 22.